# Effect of Electronic Activity Monitors and Pedometers on Health: Results from the TAME Health Pilot Randomized Pragmatic Trial

**DOI:** 10.3390/ijerph17186800

**Published:** 2020-09-18

**Authors:** Zakkoyya H. Lewis, Kenneth J. Ottenbacher, Steve R. Fisher, Kristofer Jennings, Arleen F. Brown, Maria C. Swartz, Eloisa Martinez, Elizabeth J. Lyons

**Affiliations:** 1College of Science, Department of Kinesiology and Health Promotion, California State Polytechnic University Pomona, 3801 West Temple Ave., Pomona, CA 91768, USA; 2School of Health Professions, Division of Rehabilitation Sciences, University of Texas Medical Branch, 301 University Blvd., Galveston, TX 77555, USA; kottenba@utmb.edu; 3School of Health Professions, Department of Physical Therapy, University of Texas Medical Branch, 301 University Blvd., Galveston, TX 77555, USA; stfisher@utmb.edu; 4Department of Biostatistics, MD Anderson Cancer Center, 1400 Pressler St., Unit 1411, Houston, TX 77030-4008, USA; KJennings@mdanderson.org; 5School of Medicine, Division of General Internal Medicine and Health Services Research, University of California Los Angeles, 1100 Glendon, Ave., Los Angeles, CA 90095, USA; abrown@mednet.ucla.edu; 6Department of Pediatrics, Division of Pediatrics, MD Anderson Cancer Center, 7777 Knight Rd., Houston, TX 77054, USA; MChang1@mdanderson.org; 7Sealy Center on Aging, University of Texas Medical Branch, 301 University Blvd., Galveston, TX 77555, USA; esmartin@utmb.edu; 8School of Health Professions, Department of Nutrition and Metabolism, University of Texas Medical Branch, 301 University Blvd., Galveston, TX 77555, USA; ellyons@utmb.edu

**Keywords:** cardiovascular, older adults, activity monitor, primary care, physical activity

## Abstract

**Background:** Brief counseling and self-monitoring with a pedometer are common practice within primary care for physical activity promotion. It is unknown how high-tech electronic activity monitors compare to pedometers within this setting. This study aimed to investigate the outcomes, through effect size estimation, of an electronic activity monitor-based intervention to increase physical activity and decrease cardiovascular disease risk. **Method:** The pilot randomized controlled trial was pre-registered online at clinicaltrials.gov (NCT02554435). Forty overweight, sedentary participants 55–74 years of age were randomized to wear a pedometer or an electronic activity monitor for 12 weeks. Physical activity was measured objectively for 7 days at baseline and follow-up by a SenseWear monitor and cardiovascular disease risk was estimated by the Framingham risk calculator. **Results:** Effect sizes for behavioral and health outcomes ranged from small to medium. While these effect sizes were favorable to the intervention group for physical activity (PA) (d = 0.78) and general health (d = 0.39), they were not favorable for measures. **Conclusion:** The results of this pilot trial show promise for this low-intensity intervention strategy, but large-scale trials are needed to test its efficacy.

## 1. Introduction

Cardiovascular disease is the leading cause of death world-wide [1]; however, approximately 12% of cardiovascular disease related deaths are attributed to physical inactivity [2]. For primary and secondary prevention of cardiovascular disease it is recommended that individuals take part in at least 150 min of moderate intensity physical activity (PA) a week. For older adults, who have an increased risk for cardiovascular disease [3], this recommendation equates to 7000 to 10,000 steps per day [4]. Unfortunately, older adults fall far below this recommendation [5]. Self-monitoring of behavior is an effective behavioral strategy to increase PA among inactive individuals [6]. 

Electronic activity monitors (EAMs) are commercially-available technologies that are recommended to self-monitor behavior [7]. EAMs are operationally defined as “a wearable device that objectively measures lifestyle PA and can provide feedback, beyond the display of basic activity count information, via the monitor display or through a partnering application to elicit continual self-monitoring of activity behavior [8]”. EAMs are growing in popularity, with approximately 3.3 million units sold in 2014 [9]. They provide an adequate estimation of PA [10] and they are proliferating in community-based PA interventions [8]. In addition to self-monitoring of behavior, some EAM devices offer other behavior change techniques such as: providing feedback, goal-setting, planning, social support, social comparisons, commitment, instructions on how to perform a behavior, and information on consequences [11]. The implementation of these behavior change techniques is meaningful because these strategies are known to significantly improve PA [12]. There is evidence that EAMs can increase PA and improve cardiovascular disease related outcomes [8] but evidence is lacking on their of their effectiveness in a primary care setting [7].

PA interventions through primary care are common in cardiovascular disease prevention for they rely on the strong clinician–patient relationships and the longitudinal nature of primary care [13]. It is recommended that these interventions take a two-tiered approach to promote PA incorporating both brief behavioral counseling and technology-based resources [14]. Pedometers can be used as a technology resource to facilitate self-monitoring of PA [15]. Pedometers are low tech activity monitors that can significantly increase and maintain an individual’s level of PA [15]. Despite their frequent utilization in primary care-based interventions, pedometers have several limitations. Pedometers have been scrutinized for their inaccuracy [16] and their limited methods for motivating exercise [17]. Furthermore, they do not provide features that are central to preventing cardiovascular disease such as providing education and customizability [18]. For these reasons, EAMs may be more successful for primary care interventions.

EAMs are attractive in primary care because they offer the convenience of pedometers while having a potentially higher effectiveness by addressing their reported limitations. EAMs yield a high validity for measuring steps [19,20] and the embedded behavior change techniques can theoretically augment traditional behavioral counseling, and support autonomous motivation for PA [21]. Even with a modest effect size, the potential reach of EAMs coupled with brief counseling in primary care could produce a large public health impact. Therefore, we conducted the TAME health (Testing Activity Monitors’ Effect on health) pilot randomized controlled trial which aimed to investigate a low-intensity intervention to increase PA and decrease cardiovascular disease risk within the primary care setting. This study was designed to test initial feasibility. The effect sizes collected can be used to assist in development of larger trials to evaluate the intervention’s efficacy in improving PA behavior through a combination of EAM and brief counseling. We hypothesized that individuals in the EAM group would demonstrate greater improvements in PA and cardiovascular disease risk than the pedometer group. We also hypothesized that the EAM group would have greater improvements in secondary outcomes than the pedometer group. While initial feasibility results and PA change scores have been reported [22], here we focus on effect size information for the quantitative PA, function, and quality of life outcomes investigated in the study.

## 2. Materials and Methods

The methodology of TAME health is described succinctly below. Further details on methods have been previously published [21]. This study was approved by the University’s Institutional Review Board and is registered on clinicaltrials.gov (NCT02554435). This study was conducted in accordance with the Declaration of Helsinki and follows CONSORT reporting guidelines (Appendix A). 

### 2.1. Sample

Older primary care patients (N = 40) were recruited to participate in the 12-week TAME health pilot randomized controlled trial. Patients aged 55–74 years with a body mass index of 25–35 kg/m^2^, fewer than 60 min of planned exercise a week, access to a smart device and in good health were eligible. Participants were recruited in person or through posted flyers at two clinics affiliated with a large university-based health care system. Recruitment was conducted from October 2015 to June 2016. Screening for eligibility was conducted in person and over the phone. Once an individual was deemed eligible, informed consent was promptly obtained and assessment visits were scheduled. 

### 2.2. Intervention

All participants received brief 5 A’s counseling which is optimized for primary care [23]. The counseling components included: assess, advise, agree, assist, and arrange. During the counseling a researcher with a background in exercise physiology and training in motivational interviewing reviewed the participant’s PA levels, agreed on step goals, and taught behavioral change strategies. After counseling, the researcher provided an exercise prescription and randomized the participant. Due to the nature of the intervention, the participants and the assessor were not blinded to group assignment after randomization.

Participants were randomized to one of the two groups: pedometer or EAM group. Participants in the pedometer group were given a digital pedometer (Digi-walker CW-700/701, YAMAX, San Antonio, TX, USA) and a PA log to record their daily steps, activity time, and distance walked. Participants in the EAM group were given an UP24 monitor by Jawbone (San Francisco, CA, USA) and downloaded the corresponding UP app on their personal smartphones. The UP system offered an array of behavioral change techniques including: goal setting on behavior and health outcome, providing instructions and information on consequences, as well as facilitating social support [11]. More detailed information on the specific features of the UP system is available elsewhere [21]. 

### 2.3. Measures

The study consisted of two assessments conducted at baseline and at 12 weeks. All assessments were conducted at the two clinic locations. The primary outcome variables of interest were cardiovascular disease risk and PA. Cardiovascular disease risk was measured using the Framingham non-laboratory risk equation [24] and from fitness measured by the six minute walk test [25]. Variables used in the Framingham equation included sex, age, treatment of hypertension (yes or no), smoking status (yes or no), diabetes diagnosis (yes or no), systolic blood pressure, and body mass index. PA was measured across a 7-day period prior to each assessment using a Sense Wear Armband (BodyMedia, Pittsburgh, PA, USA) [26]. 

Secondary outcome variables included: anthropometrics, body composition, blood pressure, resting pulse, health status and quality of life, and physical function. Anthropometric measurements included height (cm), weight (kg), and body mass index (kg/m^2^) using a portable stadiometer (Seca Corp., Hamburg, Germany) and a portable, calibrated electronic scale (Tanita, Arlington Heights, IL, USA). Body composition was estimated by measuring waist and hip circumference (cm) and calculating the waist-to-hip ratio. Blood pressure and resting pulse was taken using a portable sphygmomanometer (Omron BP742N, Lake Forest, IL, USA). Physical function was measured objectively through a repeated chair stand and balance test, as defined by the Short Physical Performance Battery [27], and an 8-feet up-and-go test, as outlined in the Senior Fitness Test [28]. Due to the known ceiling effect in generally healthy adults [29], the gait speed assessment of the Short Physical Performance Battery was replaced with the validated 8 feet up-and-go [28]. The same equipment for objective measurement of outcomes was used at both clinical sites. 

The remaining outcomes were assessed through self-reported questionnaires. Health status and quality of life were estimated from the Medical Outcomes Study Questionnaire Short Form 36 (SF-36) [30]. Physical function was estimated from the Patient-Reported Outcomes Measurement Information System (PROMIS) Short Form v1.2-Physical function 8b [31]. 

### 2.4. Statistical Analysis

TAME health was primarily designed to test the feasibility of the intervention and estimate effect sizes; therefore, the analyses described in this paper are exploratory and no pre-specified power calculation was performed. Effect sizes were calculated from the mean change in study variables and were categorized as small (≤0.2), medium (0.5), and large (≥0.8) using Cohen’s classification [32]. The Statistical Package for the Social Sciences (IBM-SPSS, version 20) was used and the α-level was set at 0.05. Analyses were conducted using the intent-to-treat principle by carrying baseline information to follow-up. Findings related to feasibility, acceptability, and change in PA have been previously reported [22].

Descriptive analyses were conducted using means, medians, and frequencies of all study variables. Group differences at baseline were examined using independent samples t-tests for continuous variables and through Chi-Square tests for frequency variables. Little’s Missing Completely at Random test was performed to determine whether outcome data were missing at random [33]. The distribution of study variables was tested with the Kolmogorov–Smirnov and Shapiro–Wilk tests of normality. Post-intervention differences between groups were assessed using analysis of covariance (ANCOVA) for normally distributed variables and with Mann–Whitney U for non-normal data. Covariates in the analysis were baseline values of the dependent variable and any variables significantly different between groups at baseline. Analyses on the primary outcome variables (cardiovascular disease risk, fitness, PA) were conducted by a blinded statistician. Following standard protocol, only days with a minimum of 10 h of wear time from the SenseWear armband were included in the analysis [5]. Although PA goals were set in terms of steps per day, only PA minutes were included in the analysis because the SenseWear armband is not a validated measure for steps [26]. 

## 3. Results

The CONSORT flow diagram is available in Figure 1. At baseline, the mean age and body mass index of the participants was 63.6 ± 5.3 years and 30.3 ± 3.1 kg/m^2^, respectively. A total of 75% of the participants were female, 65% were non-Hispanic White, and 55% graduated college. The demographic information by study group is available in Table 1. The mean heart/vascular age of the participants were approximately 74 ± 11.2 years with a Framingham risk score of 18.9%. Participants averaged 31.3 ± 29.4 min of moderate-vigorous PA per day and 4204.8 ± 2199.8 steps per day. Groups only differed at baseline in systolic blood pressure. Participants were comparable on all other study variables. Characteristics were not different by clinical location. Participants that did not complete the study were not significantly different on the tested variables. However, the relationship between group and missingness was significant (*p* = 0.04). The odds ratio for missingness was 0.103 (0.002, 0.956) which signifies that missingness was more likely in the pedometer group. There were no adverse events related to the intervention. 

### 3.1. Primary Outcomes

Baseline, follow-up values and the estimated effect size using the intent-to-treat principle for all study variables are outlined in Table 2. Primary outcomes were considered normally distributed (*p* > 0.05) and group differences were assessed with ANCOVA, controlling for baseline values and systolic blood pressure. The EAM intervention produced a large effect on minutes of PA and a negligible effect on cardiovascular risk and fitness. 

### 3.2. Secondary Outcomes

Functional measures, health status, and heart age required the use of nonparametric tests. All other secondary outcomes were analyzed using ANCOVA, controlling for baseline values and systolic blood pressure. As anticipated, there were no significant group differences at 12-weeks on any secondary outcomes (Table 2). The EAM intervention produced a medium effect on waist-to-hip ratio and tandem balance time. The intervention produced a negative medium effect for chair stand and 8 feet up and go time. The EAM intervention also produced a small-to-medium effect on weight and SF-36 sub-scales, with the exception of social functioning and pain.

## 4. Discussion

This analysis of a pilot randomized controlled trial aimed to investigate effect sizes for behavioral and health-related outcomes, of an EAM-based intervention for decreasing cardiovascular disease risk, increasing PA, and improving secondary outcomes. Findings suggested that both the EAM and pedometer groups increased their fitness and increased their minutes of PA. Due to the pilot nature of this study, statistical significance for these outcomes should be viewed with caution. However, the magnitude of change, which indicates potential clinical significance, was greater in the EAM group. 

A previous pilot trial evaluation by Cadmus-Bertram et al. of an EAM also found no group differences among post-menopausal women [34]. Participants were randomized to receive a Fitbit One EAM or a pedometer and were encouraged to become more physically active. After 16-weeks the EAM group increased their PA by 62 minutes while the pedometer group increased by 13 min. Although the magnitude of change was greater in those that wore a Fitbit, the differences were not significant between groups in the small study [34], likely due to low power inherent to pilot studies. The analyses from our TAME health study using a Jawbone EAM were similar. These preliminary results may suggest that, regardless of the type of monitor used, self-monitoring behaviors in combination with brief counseling can increase PA among older adults. This concept aligns with the literature of clinic-based and technology-based PA interventions [35,36,37]. 

The increased minutes of moderate and vigorous PA within our EAM group is consistent with the literature. Aittasalo et al. found that providing 5As counseling in a clinic can result in a 28 min increase in moderate to vigorous PA a week after 2 months [35]. Similarly, among chronically ill patients the combination of 5As counseling administered over 4–6 months and an EAM system resulted in an 8.9 min increase in exercise per day [36]. Activity monitoring with an EAM, Fitbit, for 3 months along with PA education increased PA by 11 min per day [37]. Participants using the UP24 monitor in our study had an increase of approximately 11 min of PA a day, whereas our pedometer group had an increase of less than 1 min of PA a day. 

Effect sizes suggested that the EAM group saw greater increase in PA, however the pedometer group was greater at baseline. The difference was non-significant, but it brings up some considerations. Two participants in the EAM group started the intervention after completing physical therapy post-knee replacement. They were included in the intervention because they met the inclusion criteria at the start of their participation and their primary care physician did not advise against PA. However, both participants had extremely low PA levels at baseline which could have contributed to the lower EAM group average. One of these participants dropped out of the study while the other participant increased their PA at a similar rate to other EAM group participants. Considering the pedometer group averaged approximately 40 min of PA a day at baseline, there may have been a ceiling effect with regard to PA activity. In pilot studies with small sample sizes, the impacts of even a few individuals can be substantial. Future research should evaluate the PA capacity within this population. In addition, future research should consider blocking randomization based on baseline PA levels.

In terms of reducing cardiovascular disease risk, we found modest increases in fitness among both groups which were comparable to providing physician advice and educational material in clinic [38]. Digital health interventions have been shown to reduce the 10-year Framingham risk score by 1.24%, systolic blood pressure by 2.12 mmHg, and weight by 1.52 kg [39]. We found more conservative changes in these outcomes in both study groups. 

Our results indicated that the EAM intervention produced a positive medium effect on waist-to-hip ratio and tandem balance time but a negative medium effect for chair stand time and 8 feet up and go time. The increase in chair stand time and 8 feet up and go time among the EAM group suggest that their physical function did not improve. However, the EAM group had greater increases in physical functioning based on self-reported measures, PROMIS and SF-36. The EAM intervention produced a favorable small-to-medium effect in both of these measures. It is possible that the decline in objective physical function is the result of measurement error. Participants completed the assessments in the same location using the same equipment and instructions, but EAM participants may have been more cautious during the follow-up. However, more research is needed to investigate the effects of an EAM on physical function.

### 4.1. Limitations and Strengths 

This study has limitations. TAME health was primarily designed to test the feasibility of the intervention. To that end, there were no blinded assessors or follow up assessments to assess maintenance. In addition, the study was not powered to detect small group differences. Based on our reported effect sizes, a group difference may be observed with a larger sample. There is also possible volunteer and selection bias in the study. Participants were volunteers and may be more intrinsically motivated to exercise than other individuals of the same age. Eligible participants were required to have regular access to a smart device; therefore the results cannot be generalized to all older adults that do not own or have access to a smart device. There was a significant difference in systolic blood pressure between study groups which suggest that the randomization procedures may not have be sufficient for the sample. Lastly, this study was not designed to determine which aspects of the EAMs contribute to the resulting effect size; although, we theorized EAM use can increase intrinsic motivation for PA [21]. Future research should identify the mechanism by which EAMs produce an effect. 

The major strength of this study is that it was a comparative effectiveness evaluation. This study adds to a small body of the literature that directly compares low- and high-tech activity monitors [34]. Another strength of this study its ability to test the current recommendation for PA promotion in primary care [14,21,40]. 

### 4.2. Implications

The American Heart Association encourages healthcare providers to provide 5A’s interventions and provide technology-based resources for individuals at moderate risk for cardiovascular disease [14,40]. Our results provide preliminary evidence that adhering to this recommendation results in clinically meaningful improvements in health among older adults. Although not statistically significant, our results also suggest that EAMs produce a small-to-medium effect over a low-tech pedometer. Healthcare systems have the potential to deliver disseminable interventions that can impact the health of their priority population if they routinely administer the low-intensity, low-impact 5A’s counseling for all patients at risk and these patients regularly self-monitor their behavior [14,40]. Large-scale, multi-site trials are needed to address the limitations of the current study and to determine the intervention’s effectiveness on a population level. 

## 5. Conclusions

PA promotion in primary care through 5A’s counseling and self-monitoring is recommended for individuals at moderate risk of cardiovascular disease [14,40]. Our evaluation of self-reported sedentary, overweight adults aged 55–74 years of age suggested that a pedometer or an EAM (Jawbone UP24) can be used with 5A’s counseling to improve health, and there is a small-to-medium effect of the EAM intervention on PA and health when compared to use of a pedometer. Future research should determine if a low-intensity study like TAME health can be broadly disseminated by healthcare systems to positively impact the health of their priority population. 

## Figures and Tables

**Figure 1 ijerph-17-06800-f001:**
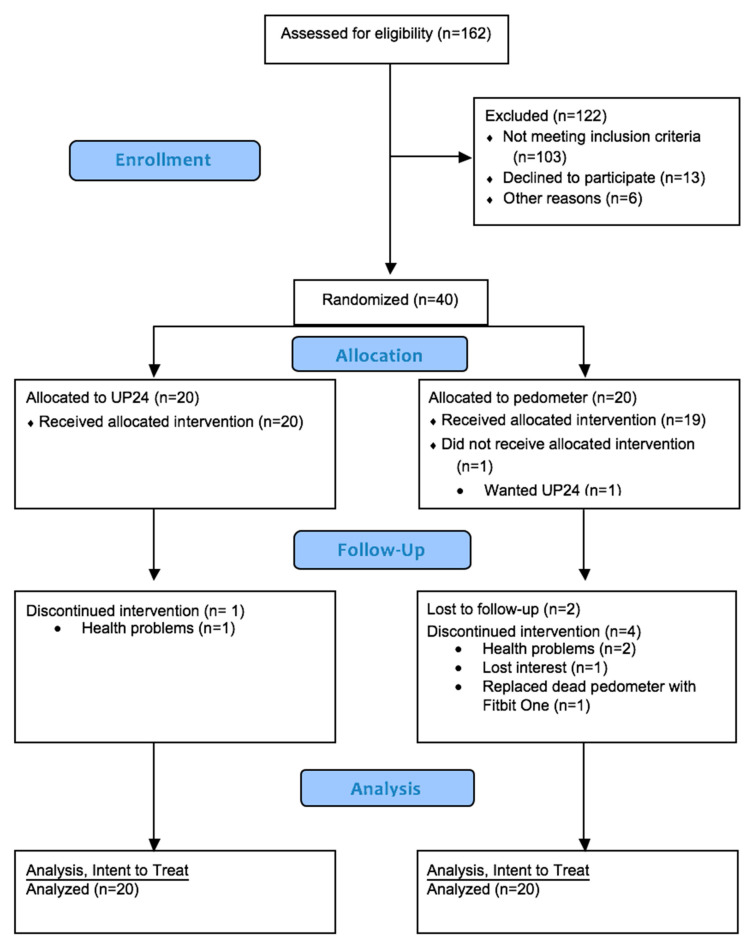
Consort flow diagram.

**Table 1 ijerph-17-06800-t001:** Demographic information by study group (*n* = 40).

	Electronic Activity Monitor	Pedometer	All
	*n* (%)		
**Age, years; mean (SD)**	64 (5.1)	63.2 (5.7)	63.6 (5.3)
**Female**	17 (85)	13 (65)	30 (75)
**Non-Hispanic White**	12 (60)	14 (70)	26 (65)
**Hispanic**	3 (15)	2 (10)	5 (12.5)
**Black/African American**	4 (20)	3 (15)	7 (17.5)
**Other**	1 (5)	1 (5)	2 (5)

**Table 2 ijerph-17-06800-t002:** Baseline, follow-up values and effect sizes for tested variables.

	Electronic Activity Monitor (*n* = 20)		Pedometer (*n* = 20)		Effect Size
	**mean (SD)/median ^†^ (IQR)**				
**Primary Outcomes**	Baseline	Follow-up	Baseline	Follow-up	
Framingham non-laboratory risk score	16.5 (12.2)	16.6 (13.2)	21.2 (13.6)	21.2 (12.6)	−0.01
Six-minute walk, ft.	1487.4 (310.3)	1561.0 (353.6)	1568.4 (354.6)	1642.9 (287.3)	−0.01
Moderate/Vigorous Activity, minutes	22.6 (21.5)	33.8 (27.6)	40.0 (33.9)	40.2 (37.5)	0.78
**Secondary Outcomes**					
^†^ Heart/Vascular age, years	71.0 (20.0)	69.0 (23.0)	79.5 (17.0)	78.0 (20.0)	0.03
Weight, kg	81.7 (10.7)	81.9 (11.2)	94.1 (40.5)	86.8 (13.6)	0.22
BMI, kg/m^2^	30.0 (3.2)	30.0 (3.5)	30.6 (3.1)	30.5 (3.1)	0.13
Waist to Hip ratio	0.8 (0.8)	0.8 (0.1)	0.9 (0.9)	0.8 (0.1)	0.45
Systolic blood pressure, mmHg *	125.0 (11.6)	125.0 (14.5)	134.8 (15.8)	134.3 (14.6)	−0.00
Diastolic blood pressure, mmHg	79.6 (10.5)	80.0 (9.8)	83.1 (9.0)	83.4 (10.0)	0.01
Resting pulse, bpm	69.4 (11.4)	69.1 (14.0)	76.3 (10.3)	76.5 (11.9)	−0.05
^†^ Chair stand, sec	13.0 (9.5)	13.3 (7.2)	14.8 (3.3)	13.8 (4.4)	0.49
^†^ Tandem balance, sec	10.0 (4.6)	10.0 (0.0)	10.0 (0.0)	10.0 (0.0)	0.46
^†^ 8 feet up and go, sec	5.9 (2.2)	6.0 (2.5)	6.0 (2.0)	5.9 (1.9)	0.49
^†^ PROMIS Physical Function	26.0 (16.0)	31.0 (11)	36.5 (11.0)	37.0 (12.0)	0.07
^†^ SF-36: Physical functioning	65.0 (50.0)	70.0 (55.0)	80.0 (26.0)	80.0 (19.0)	0.27
^†^ SF-36: Physical health role limitations	56.3 (56.3)	62.5 (56.3)	68.8 (51.6)	75.0 (46.9)	0.26
^†^ SF-36: Emotional role limitations	75.0 (58.3)	75.0 (41.7)	91.7 (62.5)	79.2 (45.8)	0.20
^†^ SF-36: Energy/fatigue	50.0 (18.8)	50.0 (25.0)	53.1 (20.3)	62.5 (20.3)	−0.31
^†^ SF-36: Emotional well-being	65.0 (30.0)	75.0 (20.0)	72.5 (18.0)	80.0 (16.0)	−0.22
^†^ SF-36: Social functioning	62.5 (50.0)	75.0 (37.5)	75.0 (56.3)	75.0 (53.1)	0.08
^†^ SF-36: Pain	45.0 (47.5)	45.0 (55.0)	62.5 (55.6)	67.5 (47.5)	−0.04
^†^ SF-36: General health	61.1 (19.4)	66.7 (19.4)	62.5 (22.6)	59.7 (26.4)	0.39

BMI: Body Mass Index; EAM: Electronic Activity Monitor; PROMIS: Patient-Reported Outcomes Measurement Information System; SF-36: Medical Outcomes Study Questionnaire Short Form 36. * significantly different between groups at baseline, *p* < 0.05. Cohen’s d for change in the tested variable between the electronic activity monitor group against the pedometer. ^†^ Variable had a non-normal distribution

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
