# Peer review of "Effect of Electronic Activity Monitors and Pedometers on Health: Results from the TAME Health Pilot Randomized Pragmatic Trial"

_ijerph, 2020, doi:10.3390/ijerph17186800_

Round 1
Reviewer 1 Report
Limitations:
This manuscript reports preliminary findings of an exploratory study designed to test the feasibility of the intervention and estimate effect sizes. The sample size was not premised on pre-specified power calculation.
The extent that arbitrary abbreviations are used detracts from the article readability.
Introduction lines 61-2: "EAMs are attractive in primary care because they have the appeal of pedometers while having a potentially higher effectiveness." The authors should elucidate why they anticipated higher potential effectiveness of EAMs cf pedometers. What distinguishing aspect of those classes of activity-monitoring devices is being eluded to?
From the conceptual rationale for this pilot study through to the discussion, the obvious question, “If EAMs produce a small-to-medium effect over a low-tech pedometer, then what is it about EAM that accounts for the difference from baseline PA by 12-weeks in PA cf pedometer group in the TAME trial?” That is not satisfactorily addressed in the introduction or discussion. To what extent could the real difference be attributed to motivational aspects of the corresponding UP app that accompanied randomization to the UP24 monitor? The difference of 62 minutes FitBit-One EAM) vs 13 minutes of PA (analogue pedometer) in the Cadmus-Bertram et. al. pilot study suggests that EAMs bundle a comparatively strong motivational component beyond counting steps per day.
strengths:
PA was operationalized on several parameters.
The Framingham equation was use to compare baseline CVD risk between groups, and risk scores were not different at 12-week follow-up.
All participants received brief 5 A’s (assess, advise, agree, assist, and arrange) counseling. During the counseling a
researcher with a background in exercise physiology and training in motivational interviewing reviewed the participant’s PA levels, agreed on step goals, and taught behavioral change strategies. After counseling, the researcher provided an exercise prescription and randomized the participant.
The study report follows CONSORT reporting guidelines.
Author Response
Reviewer 1
We appreciate the comments from Reviewer 1, particularly the recognized strengths of the manuscript. We made the following adjustments to address the limitations.
Comment 1: The extent that arbitrary abbreviations are used detracts from the article readability.
Response 1: We removed the CVD abbreviation to reduce the frequency of arbitrary abbreviations. The remaining abbreviations are electronic activity monitor (EAM) and physical activity (PA).
Comment 2: Introduction lines 61-2: "EAMs are attractive in primary care because they have the appeal of pedometers while having a potentially higher effectiveness." The authors should elucidate why they anticipated higher potential effectiveness of EAMs cf pedometers. What distinguishing aspect of those classes of activity-monitoring devices is being eluded to?
Response 2: This statement is eluding that EAMs can improve upon the limitations of pedometers such as measurement validity and the implementation of behavior change techniques. We revised the sentence to the following: EAMs are attractive in primary care because they offer the convenience of pedometers while having a potentially high effectiveness by addressing their reported limitations. EAMs yield a high validity for measuring steps [19,20] and the embedded behavior change techniques can theoretically augment traditional behavioral counseling, and support autonomous motivation for PA [21]. (Lines 63-67)
Comment 3: From the conceptual rationale for this pilot study through to the discussion, the obvious question, “If EAMs produce a small-to-medium effect over a low-tech pedometer, then what is it about EAM that accounts for the difference from baseline PA by 12-weeks in PA cf pedometer group in the TAME trial?” That is not satisfactorily addressed in the introduction or discussion. To what extent could the real difference be attributed to motivational aspects of the corresponding UP app that accompanied randomization to the UP24 monitor? The difference of 62 minutes FitBit-One EAM) vs 13 minutes of PA (analogue pedometer) in the Cadmus-Bertram et. al. pilot study suggests that EAMs bundle a comparatively strong motivational component beyond counting steps per day.
Response 3: This paper was focused on the cardiovascular and physical activity outcomes of the TAME health pilot intervention. The underlying conceptual framework of this study is based on motivation. The conceptual framework on how this study targets intrinsic motivation is described in our protocol paper (Lewis, Z.H.; Ottenbacher, K.J.; Fisher, S.R.; Jennings, K.; Brown, A.F.; Swartz, M.C.; Lyons, E.J. Testing activity monitors’ effect on health: Study protocol for a randomized controlled trial among older primary care patients. JMIR research protocols 2016, 5, e59.)
We added a brief statement to the introduction that highlights the importance of behavioral change techniques. The implementation of these behavior change techniques is meaningful because these strategies are known to significantly improve PA [12]. (Lines 48-49) In our discussion, we added a statement addressing the limitations of this study to determine the mediating relationship of motivation between electronic activity monitors and physical activity. Lastly, this study was not designed to determine which aspects of the EAMs contribute to the resulting effect size; although we theorized EAM use can increase intrinsic motivation for PA [21]. Future research should identify the mechanism in which EAMs produce an effect. (Lines 251-254)
Reviewer 2 Report
This work compared physical activity and cardiovascular disease risk between 20 subjects who wore an electronic activity monitor (EAM) and 20 subjects who wore a pedometer, during a 12-weeks program in a primary care setting. The study had the design of a randomized controlled trial, and it was described thoroughly in a previous paper (reference 18). The work is original and relevant, and the manuscript was well-written. However, there are several issues that must be addressed, as described below.
- The aim of the study was to compare the effectiveness of an EAM and a podometer to increase physical activity and to decrease cardiovascular risk, as is mentioned in the Abstract (page 1, line 16) and the Introduction (page 2, line 64). However, the study considered a very small number of participants (n = 40), and the authors justify by saying that this clinical trial was primarily designed to test the feasibility of the intervention and estimate effect sizes (page 3, line 124). It is not surprising from the small number of subjects (20 per group) and the large dispersion in the study variables that there were no significant differences between groups, and there were no significant changes during follow-up compared to baseline (i.e., the p-value was > 0.05). Considering this, the interpretation of results is very straightforward: there was no significant increase in physical activity or a significant decrease in cardiovascular risk in both groups, and the efficacy of the intervention was not different between groups. Instead, the current manuscript seems to be focused on the estimation of the effect size (for instance: Abstract, page 1, line 25; Discussion, page 7, line 180; Conclusions page 8, line 254), which was not the main aim as mentioned in the manuscript. Moreover, there are many instances in which the interpretation of results is incorrect, for instance: “…Both groups increased their fitness and increased their minutes of PA” (page 7, line 178); “the increased minutes of moderate and vigorous PA within our EAM group is consistent with the literature” (page 7, line 191); “The EAM group did have a greater increase in PA...” (page 7, line 199); “In terms of reducing CVD risk, we found modest increases in fitness among both groups” (page 7, line 209); “…the EAM group had more favorable changes in CVD risk factors…” (page 7, line 213); “Our results provide preliminary evidence that adhering to this recommendation results in clinically meaningful improvements in health among older adults” (page 8, line 242). The authors should carefully update the interpretation of the results throughout the manuscript.
- The Conclusions section should mention only statements that are based on the results of the present study, and it should avoid speculation. Phrases such as: “PA promotion in primary care through 5A’s counseling and self-monitoring is recommended for individuals at moderate risk for CVD” and “Because of its low-intensity and highly scalable nature, this intervention has the potential to be broadly disseminated by healthcare systems to positively impact the health of their priority population” are not conclusions of the present work.
- The statistical analysis section should mention the test used to verify if the variables had a normal distribution. The variables that did not have variable distribution should be described as median (percentile 25 – percentile 75). The data reported in several variables (for instance minutes of moderate/vigorous activity) shows very large dispersion (for instance minutes of activity with mean = 22.6 and standard deviation = 21.5), which is likely due to non-normal distribution. It is very important to update the descriptive and inferential analysis to an appropriate statistical method (parametric or non-parametric). This is also extended to the estimation of the effect size.
- One result that is particularly highlighted by the authors is the large effect size calculated for moderate/vigorous activity in minutes. It is remarkable that one of the samples (pedometer group, follow-up) had a dispersion (standard deviation = 3.7) that was one order of magnitude smaller than the other samples (standard deviation between 21.5 and 33.9). Why is that? In the discussion section (page 7, line 204) the authors mention: “Considering the pedometer group averaged approximately 40 minutes of PA a day at baseline, there may have been a ceiling effect in regard to PA activity”. It seems that such a supposed ceiling effect is trying to explain the lack of change in the mean physical activity in this group, and it has nothing to do with the unusually low dispersion in the sample during follow-up. It is strongly recommended to double-check the data and the variable distribution in that subgroup.
- There are clear differences between groups in several variables during the baseline evaluation, including systolic blood pressure and minutes of moderate/vigorous activity (Table 2). This suggests some bias in the group assignments, which may compromise the randomized allocation of the participants. How do the authors explain that? The authors explain a potential bias regarding the baseline physical activity in the EAM group due to the inclusion of two participants that started the intervention after completing physical therapy post-knee replacement (page 7, line 199). Why was this not considered an exclusion criterion?
Author Response
Reviewer 2
We appreciate the thoughtful comments from Reviewer 2. We believe that addressing these comments have improved manuscript.
Comment 1: The aim of the study was to compare the effectiveness of an EAM and a podometer to increase physical activity and to decrease cardiovascular risk, as is mentioned in the Abstract (page 1, line 16) and the Introduction (page 2, line 64). However, the study considered a very small number of participants (n = 40), and the authors justify by saying that this clinical trial was primarily designed to test the feasibility of the intervention and estimate effect sizes (page 3, line 124). It is not surprising from the small number of subjects (20 per group) and the large dispersion in the study variables that there were no significant differences between groups, and there were no significant changes during follow-up compared to baseline (i.e., the p-value was > 0.05). Considering this, the interpretation of results is very straightforward: there was no significant increase in physical activity or a significant decrease in cardiovascular risk in both groups, and the efficacy of the intervention was not different between groups. Instead, the current manuscript seems to be focused on the estimation of the effect size (for instance: Abstract, page 1, line 25; Discussion, page 7, line 180; Conclusions page 8, line 254), which was not the main aim as mentioned in the manuscript. Moreover, there are many instances in which the interpretation of results is incorrect, for instance: “…Both groups increased their fitness and increased their minutes of PA” (page 7, line 178); “the increased minutes of moderate and vigorous PA within our EAM group is consistent with the literature” (page 7, line 191); “The EAM group did have a greater increase in PA...” (page 7, line 199); “In terms of reducing CVD risk, we found modest increases in fitness among both groups” (page 7, line 209); “…the EAM group had more favorable changes in CVD risk factors…” (page 7, line 213); “Our results provide preliminary evidence that adhering to this recommendation results in clinically meaningful improvements in health among older adults” (page 8, line 242). The authors should carefully update the interpretation of the results throughout the manuscript.
Response 1: The true aims of this project were described in the published protocol paper (Lewis, Z.H.; Ottenbacher, K.J.; Fisher, S.R.; Jennings, K.; Brown, A.F.; Swartz, M.C.; Lyons, E.J. Testing activity monitors’ effect on health: Study protocol for a randomized controlled trial among older primary care patients. JMIR research protocols 2016, 5, e59.) The specific aims of the project were:
AIM 1: Evaluate the feasibility and acceptability of implementing a technology-enhanced brief intervention to increase PA in a primary care setting. Measures of feasibility included days the EAM was worn, usage of the app, technological problems, attrition, self-regulation and adverse events. Acceptability was measured by self-report and focus groups.
AIM 2a: Compare the counseling plus EAM intervention to a counseling plus pedometer intervention. Primary outcomes were changes in PA and cardiovascular risk. We also investigated secondary outcomes (differences in adherence, weight and body composition, health status, physical function). The conceptual framework for the intervention is shown in Figure 1-2 and Table 1.
AIM 2b: Compare the counseling plus EAM intervention to a counseling plus pedometer intervention on changes in Self-Determination Theory construct variables. These variables include autonomous regulation and basic psychological need fulfillment.
The specific aim of this paper was to address Aim 2a; however, due to the pilot nature of the intervention efficacy could not be determined. We recognize that the small sample size yields large dispersions and lacks statistical significance. We have updated the language throughout the manuscript to emphasize that this project was meant to garner effect sizes that could be used to power an efficacious intervention. We also removed the reported p-values of primary outcomes as they are not interpretable; only the effect sizes are reported.
The initial submission did not describe the published results of Aim 1 regarding the feasibility and acceptability outcomes of the intervention. We added the following to emphasize the aim of this paper: While initial feasibility results and PA change scores have been reported [22], here we focus on effect size information for the quantitative PA, function, and quality of life outcomes investigated in the study. (Lines 76-78).
Comment 2: The Conclusions section should mention only statements that are based on the results of the present study, and it should avoid speculation. Phrases such as: “PA promotion in primary care through 5A’s counseling and self-monitoring is recommended for individuals at moderate risk for CVD” and “Because of its low-intensity and highly scalable nature, this intervention has the potential to be broadly disseminated by healthcare systems to positively impact the health of their priority population” are not conclusions of the present work.
Response 2: We agree that we need to provide clarity to our conclusions to ensure that they mirror the results. The first phrase referenced by the reviewer is in fact from a reference and the citation is now included. The second phrase now reads “Future research should determine if a low-intensity study like TAME health can be broadly disseminated by healthcare systems to positively impact the health of their priority population. (Lines 275-277)”
We also revised the language in the discussion to reflect the that this study was designed to test feasibility and the impact on PA cannot be determined (Lines 188-193).
Comment 3: The statistical analysis section should mention the test used to verify if the variables had a normal distribution. The variables that did not have variable distribution should be described as median (percentile 25 – percentile 75). The data reported in several variables (for instance minutes of moderate/vigorous activity) shows very large dispersion (for instance minutes of activity with mean = 22.6 and standard deviation = 21.5), which is likely due to non-normal distribution. It is very important to update the descriptive and inferential analysis to an appropriate statistical method (parametric or non-parametric). This is also extended to the estimation of the effect size.
Response 3: Description of the normality test was originally omitted from this paper because the complete statistical plan was described in the published protocol paper. However, we agree with the reviewer that the brief description is necessary. We added the following sentence “The distribution of study variables was tested with the Kolmogorov-Smirnov and Shapiro-Wilk tests of normality. (Lines 146-147)” Although it was implied in the methods section that variables with a non-normal distribution were analyzed with non-parametric analyses, we emphasized within Table 2 which variables were analyzed using these procedures. This was also added to the text within the results section. “Primary outcomes were considered normally distributed (p>0.05) and group differences were assessed with ANCOVA… (Lines 174-176)” In addition, we added the median and IQR for all non-normally distributed variables in Table 2.
Comment 4: One result that is particularly highlighted by the authors is the large effect size calculated for moderate/vigorous activity in minutes. It is remarkable that one of the samples (pedometer group, follow-up) had a dispersion (standard deviation = 3.7) that was one order of magnitude smaller than the other samples (standard deviation between 21.5 and 33.9). Why is that? In the discussion section (page 7, line 204) the authors mention: “Considering the pedometer group averaged approximately 40 minutes of PA a day at baseline, there may have been a ceiling effect in regard to PA activity”. It seems that such a supposed ceiling effect is trying to explain the lack of change in the mean physical activity in this group, and it has nothing to do with the unusually low dispersion in the sample during follow-up. It is strongly recommended to double-check the data and the variable distribution in that subgroup.
Response 4: We double-checked the data and found that the standard deviation for the pedometer group at follow-up was reported incorrectly. Thank you for the meticulous review of the manuscript. All of the data presented has been double checked and the accurate values are entered in Table 2.
Comment 5: There are clear differences between groups in several variables during the baseline evaluation, including systolic blood pressure and minutes of moderate/vigorous activity (Table 2). This suggests some bias in the group assignments, which may compromise the randomized allocation of the participants. How do the authors explain that? The authors explain a potential bias regarding the baseline physical activity in the EAM group due to the inclusion of two participants that started the intervention after completing physical therapy post-knee replacement (page 7, line 199). Why was this not considered an exclusion criterion?
Response 5: We recognize that the randomization strategy may not have been optimal for this sample size. We added the following sentence to our limitation section: “There was a significant difference in systolic blood pressure between study groups which suggest that the randomization procedures may not have be sufficient for the small sample. (Lines 249-251)” In reference to the two participants that were included after completing physical therapy, they met the inclusion criteria for the study and their participation was warranted based on our exclusion criteria:
Individuals will be excluded from the study if participation in PA is inadvisable by their doctor, they are involved in another PA intervention currently or within the past 6 months, utilized an EAM in the past 6 months, are unwilling to travel for scheduled visits, currently taking medications that affect body composition, a current smoker, report of alcohol or drug problem, institutionalizations for psychiatric illness within the last year, or do not consent. (Lewis, Z.H.; Ottenbacher, K.J.; Fisher, S.R.; Jennings, K.; Brown, A.F.; Swartz, M.C.; Lyons, E.J. Testing activity monitors’ effect on health: Study protocol for a randomized controlled trial among older primary care patients. JMIR research protocols 2016, 5, e59.)
We revised the section in the discussion to now read “They were included in the intervention because they met the inclusion criteria at the start of their participation and their primary care physician did not advise against PA. However, both participants had extremely low physical activity levels at baseline which could have contributed to the lower EAM group average. (Lines 216-219).

Round 2
Reviewer 2 Report
The manuscript improved substantially. There are no further comments.